# Association between Dietary Cholesterol and Their Food Sources and Risk for Hypercholesterolemia: The 2012–2016 Korea National Health and Nutrition Examination Survey

**DOI:** 10.3390/nu11040846

**Published:** 2019-04-15

**Authors:** Dongjoo Cha, Yongsoon Park

**Affiliations:** Department of Food and Nutrition, Hanyang University, 222 Wangsimni-ro, Seongdong-gu, Seoul 04763, Korea; djcha92@naver.com

**Keywords:** dietary cholesterol, egg, hypercholesterolemia, processed meat, saturated fatty acid

## Abstract

It remains unclear whether cholesterol intake can increase serum cholesterol. Therefore, the present study aimed to investigate the hypothesis that the risk for hypercholesterolemia was not associated with intake of dietary cholesterol after adjusting for saturated fatty acid (SFA). Based on the data from the 2012–2016 KNHANES, dietary cholesterol was positively associated with the risk for abnormalities in total cholesterol (TC) (odds ratio (OR): 1.153, 95% confidence interval (CI): 0.995–1.337; *p* = 0.028) and low-density lipoprotein cholesterol (LDL-C) (OR: 1.186, 95% CI: 1.019–1.382; *p* = 0.018) levels before adjusting for SFA; after adjusting for SFA, no significant associations were found between these variables. The mediation analysis showed that dietary cholesterol had no direct effects on the serum levels of TC and LDL-C; in contrast, SFA had significant indirect effects on the association between dietary cholesterol and serum levels of TC and LDL-C. Furthermore, processed meats, but not eggs and other meats, were positively associated with the risk for abnormalities in both TC (OR: 1.220, 95% CI: 1.083–1.374; *p* = 0.001) and LDL-C (OR: 1.193, 95% CI: 1.052–1.354; *p* = 0.004) levels. The present study suggested that higher intake of processed meats with high SFA, but not dietary cholesterol was associated with higher risk for abnormalities in TC and LDL-C levels.

## 1. Introduction

Hypercholesterolemia, particularly elevated serum levels of total cholesterol (TC), and low-density lipoprotein-cholesterol (LDL-C) have a strong and graded positive association with the risk of cardiovascular disease (CVD) [1]. Cardiovascular disease was the second leading cause of death in Korea [2], and the prevalence of hypercholesterolemia increased 2.5 times from 2005 to 2016 among Koreans aged 30 years or older [3].

There has been a long-running debate on whether serum cholesterol levels are responsive to high intake of cholesterol [4]. Previous epidemiologic studies inconsistently reported that dietary cholesterol was positively associated with serum levels of TC and LDL-C in Chinese [5,6], but not significantly associated with LDL-C in Americans [7]. In the previous studies reporting a positive association between cholesterol intake and serum levels of cholesterol, saturated fatty acid (SFA) was not adjusted as a confounding factor [5,6]; in the study reporting a negative association, SFA was adjusted [7]. Most foods with high cholesterol content are also rich in animal-based SFA, the major determinant of serum cholesterol levels [8]. Previous studies suggested that the effect of SFA on serum levels of TC and LDL-C was greater than that of dietary cholesterol itself [9,10]. Eggs and meats are the major foods that contain dietary cholesterol, but meats (not egg) also contain SFA [11]. Particularly, processed meats have higher SFA than red meats because of their high contents of visible fat [12,13]. Epidemiological studies consistently reported that eggs were not significantly associated with the increase in serum levels of TC and LDL-C in Korean [14] and Spanish populations [15]. In addition, the majority of clinical trials in general population showed that additional egg consumption did not significantly increase serum levels of TC and LDL-C [16,17,18,19,20]. Even in a recent long-term study, the consumption of 12 eggs per week for 1 year did not significantly alter the serum levels of TC and LDL-C [21].

Although only a limited number of epidemiological studies have assessed the association between meat consumption and risk for hypercholesterolemia, a meta-analysis of clinical trials showed that intake of red meat mainly from unprocessed meat had no adverse effect on serum TC and LDL-C [22]. Another meta-analysis of clinical trials suggested that consumption of red meat and white meat reduced the serum levels of TC and LDL-C [23]. It is well known that processed meat had a stronger positive association with the risk of CVD as compared with unprocessed red meat [24,25]. However, no previous study has assessed whether consumption of processed meat has adverse effect on the risk of hypercholesterolemia. The previous epidemiologic study among Koreans reported that serum levels of LDL-C were not significantly different according to the quartiles of cholesterol intake [26]. Therefore, the present study aimed to investigate the hypothesis that the risk for hypercholesterolemia was not associated with dietary intake of cholesterol after adjusting for SFA, but was positively associated with consumption of processed meat.

## 2. Materials and Methods

### 2.1. Study Population

This study was based on the data from the 2012–2016 Korea National Health and Nutrition Examination Survey (KNHANES). Participants were recruited using a clustered, multistage, stratified, and probability sampling design by age group, gender, and geographical region to represent the non-institutionalized civilian South Korean population [27]. KNHANES is composed of three component surveys: health interview, health examination, and nutrition survey. The health interview and health examination at Mobile Examination Centers were conducted by trained staff members followed by in-person household interview by dietitians for the nutrition survey. The KNHANES protocol was approved by the Institutional Review Board of Korea Centers for Disease Control and Prevention and Hanyang University (HYI-18-226); informed consent was obtained from all participants. 

A total of 39,156 participants were recruited in the study: 8058 in 2012, 8018 in 2013, 7550 in 2014, 7380 in 2015, and 8150 in 2016; however, only 11,313 were included in the present study (Appendix A). We excluded participants younger than 30 years old, who were unable to complete the demographic and dietary survey, were taking lipid-lowering drugs, and reported energy intakes less than 500 kcal/day or greater than 4000 kcal/day. 

### 2.2. Health Examination Survey

A health interview questionnaire was used to obtain data on age, gender, education level, household income, smoking status, alcohol intake, and physical activity. Physical activity was defined as walking for more than 30 min at a time at least 5 times per week or strength exercises at least 2 times per week. Body weight and height were measured using standardized techniques and calibrated equipment. Using fasting blood samples, TC and LDL-C were measured directly based on the enzymatic method using a Hitachi Automatic Analyzer 7600 (Hitachi, Tokyo, Japan). LDL-C was calculated using Friedewald’s equation for participants with no data on LDL-C [28]. TC ≥200 mg/dL (≥5.17 mmol/L) or LDL-C ≥130 mg/dL (≥3.36 mmol/L) was defined as abnormal based on the values of “borderline high” classification from the Korean Society of Lipid and Atherosclerosis criteria [29].

### 2.3. Dietary Intake Measurement

Dietary intake was assessed by 112-item dish-based semi-quantitative food frequency questionnaires (FFQ) that were developed and validated for KNHANES [30]. The frequency of consumption was divided into nine categories: never, once a month, 2–3 times a month, once a week, 2–4 times a week, 5–6 times a week, once a day, twice a day, and three or more times a day. Portions were categorized as being one of three sizes depending on the type of food: 0.5 portion, 1.0 portion, and 1.5 (or 2.0) portion. Food consumption was estimated by multiplying the frequency of food intake by the amount of servings consumed for each food item. The median values of each category of food consumption frequency were used to calculate the frequency of food intake (“never” = 0, “once a month” = 1/4.3, “2–3 times a month” = 2.5/4.3, “once a week” = 1, “2–4 times a week” = 3, “5–6 times a week” = 5.5, “7 times a week” = 7, “14 times a week” = 14, and “21 times a week” = 21). The amount of servings consumed was converted to the ratio in proportion to the standard serving size. Groups of total red, processed, and white meat were classified as described [31].

### 2.4. Statistical Analysis

All statistical analyses were performed using complex sample survey data analysis in SPSS version 24.0 (SPSS Inc., Chicago, IL, USA) in accordance with the KNHANES complex sampling design. Sample weights obtained from the KNHANES were used to achieve unbiased estimates of means and frequencies that were nationally representative of the Korean population [32]. Continuous variables were expressed as means ± standard error of the mean, and categorical variables were expressed as frequencies and percentages. The characteristics and risk factors for serum cholesterol abnormalities in study groups were compared using Student’s *t*-test or chi-square tests for continuous or categorical variables, respectively. Cholesterol intake and food consumption were categorized into tertiles based on the distribution of all participants; participants with the lowest tertile of cholesterol intake and food consumption served as the reference group. Multivariate logistic regression models were used to examine the associations between cholesterol intake and food consumption and risk for serum cholesterol abnormalities. In the multivariate models, the covariates showing a *p* value < 0.20 were selected as confounding factors and included in the fully adjusted model [33]. The *p* value for trend was calculated using multivariate logistic regression analyses by handling the median value of each category of cholesterol intake and tertiles of food consumption as a continuous value. To test whether the relation between cholesterol intake and serum levels of cholesterol was mediated by SFA intake, mediation analysis was performed using the Hayes PROCESS macro (model 4) to test the total, direct, and indirect effects of dietary cholesterol on serum levels of TC and LDL-C [34]. The significance of the mediated effect was evaluated by calculating the bias-corrected bootstrap 95% confidence intervals (95% CI). If the 95% CI did not include zero, criteria for mediation were met. In the case of significant indirect effects, the proportion of the total effect that was mediated was calculated using the following formula, (indirect effect/total effect) × 100. *p* values < 0.05 were considered significant.

## 3. Results

### 3.1. Characteristics of Participants

Participants with abnormal TC and LDL-C levels were significantly older, had a higher body mass index (BMI), lesser physical activity, and achieved a lower education level than those with normal TC and LDL-C (Table 1). Most of the participants with abnormal LDL-C levels were men and drink less than those with normal LDL-C.

Cholesterol intake was not significantly different between participants with normal and abnormal TC and LDL-C levels. Participants with abnormal LDL-C levels consumed significantly less total energy, protein, fat, monounsaturated fatty acid (MUFA), polyunsaturated fatty acid (PUFA), white meat, red meat, and processed meat, but more carbohydrates than those with normal LDL-C.

### 3.2. Associations between Dietary Cholesterol and Food Sources and Risk for Hypercholesterolemia

Logistic regression analysis showed that cholesterol intake was not significantly associated with the risk for abnormalities in TC and LDL-C levels before and after adjusting for all potential confounders (Table 2). However, the associations were significant after adjusting for all potential confounders except SFA, suggesting that SFA could be more important than cholesterol intake. In addition, mediation analysis showed that dietary cholesterol had no direct effects on serum levels of TC and LDL-C, but SFA contributed to significant indirect effects of 37% and 58% on the relationship between dietary cholesterol and serum levels of TC and LDL-C, respectively (Figure 1). 

Consumption of processed meat was positively associated with the risk for abnormalities in both TC and LDL-C levels after adjustment for all potential confounders (Table 3 and Table 4). The risk for abnormalities in TC and LDL-C levels was increased with consumption of >0.58 servings of processed meat. However, there was no significant association between intakes of red meat, white meat, and egg and the risk for abnormalities in TC and LDL-C levels after adjusting for all potential confounders.

## 4. Discussion

The present study showed that dietary cholesterol was positively associated with the risk for abnormalities in TC and LDL-C levels before adjusting for SFA, but not after adjusting for SFA. There were significant mediation effects of SFA on the association between dietary cholesterol and serum levels of TC and LDL-C. In addition, consumption of processed meat but not egg and other meats had adverse effects on the risk for abnormalities in TC and LDL-C levels. 

Lin et al. [7] reported that there was no adverse association between dietary cholesterol and serum level of LDL-C after adjusting for SFA within 20 years of follow-up in the Framingham offspring study. On the contrary, SFA was not adjusted in the two epidemiologic studies showing the positive association between dietary cholesterol and serum levels of TC and LDL-C among Chinese individuals [5,6]. Meta-analysis of clinical trials reported that the changes in serum levels of TC and LDL-C were more strongly associated with the change in SFA intake than dietary cholesterol [9]. Gratz et al. [35] demonstrated that lathosterol, a precursor in the biosynthesis of cholesterol, was increased significantly with a diet containing high SFA, suggesting that SFA stimulated the hepatic biosynthesis of cholesterol. However, the increase of cholesterol intake diminished the biosynthesis of cholesterol and led to higher excretion of endogenous cholesterol through the biliary tract to prevent the rise in serum cholesterol levels [36]. Furthermore, dietary cholesterol accounted for approximately 25% of serum cholesterol level in humans, while the rest is derived from biosynthesis of cholesterol [37]. 

The primary source of dietary cholesterol is egg, which contains a high level of cholesterol but low level of SFA [3]. Consistent with the present study, previous epidemiological studies reported that higher egg consumption was not associated with increased serum levels of TC and LDL-C [14,15]. Clinical trials also showed that egg consumption did not affect the serum levels of TC and LDL-C [16,17,18,19,20,21]. In addition, previous meta-analysis of clinical trials suggested that egg consumption resulted in increase in serum levels of TC and LDL-C only in hyper-responders [38]. Beynen et al. [39] demonstrated that hyper-responders may not have the ability to maintain cholesterol homeostasis by decreasing synthesis after increased intake of dietary cholesterol compared with hypo-responders. Furthermore, consumption of egg white protein lowered the serum levels of cholesterol in young women [40]. The protein component of egg white, including ovalbumin and ovotransferrin, was responsible for decreasing the serum cholesterol levels by inhibiting the micellar solubility of cholesterol in the intestines [41]. A recent pooled analysis of six prospective cohort studies reported that the associations of dietary cholesterol or egg consumption with incident CVD and mortality were not significant among American adults [42]. However, each additional 300 mg/day of dietary cholesterol consumed or each additional half an egg/day consumed was significantly associated with higher risk of incident CVD and mortality [42]. In the present study, the average intake of cholesterol was 254 mg/day, lower than the mean dietary cholesterol consumption of Americans, and only 28% of population consumed > 300 mg/day.

No previous study has investigated the association between processed meat consumption and risk for hypercholesterolemia. However, meta-analysis of prospective cohort studies and clinical trials consistently reported that the intake of processed meats but not intake of unprocessed red meats was positively associated with the risk of coronary heart disease [24]. Processed meats such as sausages contain a large amount of visible fat because of the superior taste and textural characteristic that comes from its higher SFA content [13,43]. Li et al. [13] reported that the SFA content of lean red meat is lesser (<1.5 g /100 g) than that of visible fat from red meat (>37 g SFA/100 g). Processed meats contained not only SFA but also salt, which could increase the risk of CVD [44]. Processed meats contain approximately 400% higher sodium than unprocessed meats for improvement of taste and preservation [45]. Although the role of dietary sodium on lipid metabolism was unclear, animals fed excess salt for 15 months to four years have higher serum cholesterol levels [46]. In addition, Bu et al. [47] reported that sodium intake had a significantly positive correlation with serum levels of TC and LDL-C in Koreans.

Compared with processed meats, red and white meats have lower SFA content [12,48]. In the present study, consumption of red and white meats had no significant effects on abnormalities in TC and LDL-C levels. Meta-analysis of clinical trials also reported that red meat consumption did not influence the serum levels of TC and LDL-C [22]. Approximately 50% of the fatty acids in beef are MUFA, primarily oleic acid, which has a cholesterol-lowering effect [49]. In addition, one-third of SFA in beef are stearic acids [49], which has no effect on serum cholesterol [50]. Furthermore, a meta-analysis of clinical trials reported that consumption of red or white meats reduced the serum TC and LDL-C levels [23]. Those red and white meats are mainly lean meats, which contain higher PUFA and lower SFA compared with visible fat in meat [13]. 

The major strength of this study lies in the fact that the data were gathered from a nationally representative survey throughout Korea; thus, the findings could be generalized to the Korean population. However, the present study also has a few limitations. First, the findings are not applicable to other non-Asian populations. Second, due to the cross-sectional design of the study, it was unable to identify the casual relationship between cholesterol intake and risk for serum cholesterol abnormalities. Third, although adjustments were made for various confounding factors, certain residual confounders may still remain. Fourth, trans fatty acid is one of the major determinants of the level of blood cholesterol. However, due to the missing information regarding the trans fatty acid in database of KNHANES, we could not include effect of trans fatty acid on the associations between dietary factors and serum cholesterol in the present study. Fifth, although the validated FFQ was used, FFQ is restricted to listed food items and the recall bias of participants might affect the dietary assessment. Finally, a “hyper-responder” phenotype could exist in that a participant is sensitive to a very low intake of dietary cholesterol; therefore, extreme care must be taken when providing blanket cholesterol recommendations for all individuals.

## 5. Conclusions

Intake of SFA mediated the association between dietary cholesterol and the risk for the abnormalities in TC and LDL-C levels, and consumption of processed meats containing high SFA was associated with higher risk for TC and LDL-C abnormalities in the Korean population. Further study is needed to investigate whether low processed meat consumption could reduce the risk for hypercholesterolemia in well-designed prospective cohort studies using a large population.

## Figures and Tables

**Figure 1 nutrients-11-00846-f001:**
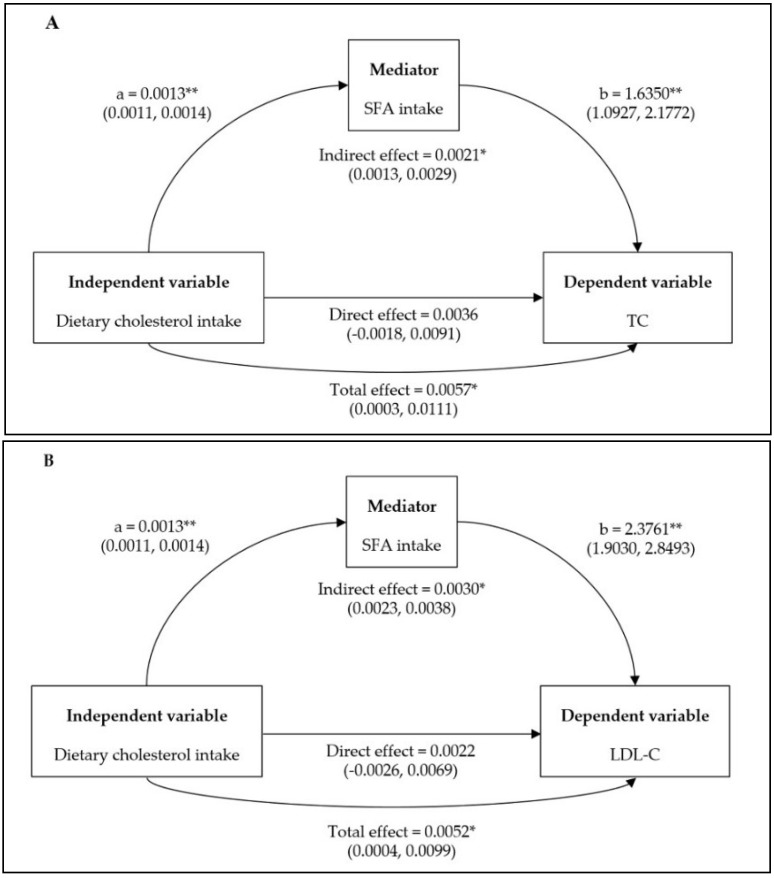
Mediation effects of saturated fatty acid (SFA) on the association between dietary cholesterol and serum levels of total cholesterol (TC; **A**), and low-density lipoprotein-cholesterol (LDL-C; **B**). Confounding factors were age, sex, BMI, physical activity, educational level, drinking, carbohydrate, and protein. Unstandardized coefficients were shown along with their estimated *p* values: “a” is the linear regression coefficient of the dietary cholesterol–SFA association; “b” is the linear regression coefficient of the SFA–serum levels of TC and LDL-C. * *p* < 0.05; ** *p* < 0.001.

**Table 1 nutrients-11-00846-t001:** Baseline characteristics of participants according to total cholesterol (TC) and low-density lipoprotein-cholesterol (LDL-C) levels ^a^.

Variables	TC		*p*-Value	LDL-C		*p*-Value
	<200 mg/dL (*n* = 6593)	≥200 mg/dL (*n* = 4720)		<130 mg/dL (*n* = 7561)	≥130 mg/dL (*n* = 3752)	
TC (mg/dL)	171.49 ± 0.30	227.56 ± 0.43	<0.001	177.97 ± 0.36	229.23 ± 0.49	<0.001
LDL-C (mg/dL)	99.98 ± 0.29	144.07 ± 0.41	<0.001	101.44 ± 0.26	152.72 ± 0.39	<0.001
Age (years)	44.38 ± 0.16	47.08 ± 0.18	<0.001	44.59 ± 0.15	47.38 ± 0.20	<0.001
Men, *n* (%)	2443 (47.5)	1754 (48.5)	0.363	2761 (46.8)	1436 (50.3)	0.002
BMI (kg/m^2^)	23.46 ± 0.05	24.55 ± 0.06	<0.001	23.52 ± 0.05	24.70 ± 0.06	<0.001
Education level, *n* (%)						
≤Elementary	643 (8.0)	644 (10.7)	<0.001	727 (7.8)	560 (11.7)	<0.001
Middle	563 (7.7)	555 (10.6)		681 (8.2)	437 (10.3)	
High	2385 (37.5)	1695 (36.7)		2774 (38.0)	1306 (35.5)	
≥College	3002 (46.8)	1826 (42.0)		3379 (45.9)	1449 (42.5)	
Household income, *n* (%)						
Low	513 (7.1)	431 (8.4)	0.081	586 (7.2)	358 (8.6)	0.136
Low-middle	1567 (24.3)	1142 (23.3)		1798 (24.0)	911 (23.4)	
Upper-middle	2158 (33.3)	1481 (32.1)		2453 (32.8)	1186 (32.7)	
High	2355 (35.4)	1666 (36.2)		2724 (36.0)	1297 (35.2)	
Smoking, *n* (%)						
Never	4245 (56.7)	2984 (54.5)	0.121	4839 (56.4)	2390 (54.5)	0.165
Former	1135 (19.8)	847 (21.4)		1312 (19.9)	670 (21.7)	
Current	1213 (23.4)	889 (24.1)		1410 (23.7)	692 (23.8)	
Drinking, *n* (%)	3681 (59.3)	2577 (60.0)	0.492	4349 (61.2)	1909 (56.3)	<0.001
Physical activity, *n* (%)	3148 (49.2)	2162 (46.5)	0.024	3605 (48.9)	1705 (46.3)	0.026
Dietary cholesterol (mg/day)	253.84 ± 2.60	254.29 ± 2.91	0.905	256.52 ± 2.46	248.99 ± 3.12	0.050
Energy (kcal/day)	2061.49 ± 11.71	2059.26 ± 13.45	0.894	2074.57 ± 11.20	2032.19 ± 15.01	0.018
Nutrients (% of energy)						
Carbohydrate	64.68 ± 0.13	64.47 ± 0.16	0.298	64.23 ± 0.12	65.33 ± 0.18	<0.001
Protein	13.00 ± 0.04	12.93 ± 0.04	0.165	13.01 ± 0.03	12.90 ± 0.04	0.034
Fat	17.23 ± 0.09	17.10 ± 0.10	0.266	17.27 ± 0.09	16.97 ± 0.11	0.019
SFA	5.00 ± 0.03	4.96 ± 0.03	0.367	5.01 ± 0.03	4.93 ± 0.04	0.069
MUFA	5.26 ± 0.03	5.22 ± 0.03	0.348	5.28 ± 0.03	5.17 ± 0.04	0.016
PUFA	4.66 ± 0.03	4.62 ± 0.03	0.238	4.67 ± 0.02	4.58 ± 0.03	0.020
Foods (servings/week)						
Egg	2.52 ± 0.03	2.54 ± 0.04	0.694	2.54 ± 0.03	2.51 ± 0.05	0.590
White meat	0.79 ± 0.01	0.77 ± 0.01	0.189	0.80 ± 0.01	0.74 ± 0.02	0.001
Red meat	3.38 ± 0.05	3.25 ± 0.01	0.050	3.40 ± 0.04	3.17 ± 0.06	0.001
Processed meat	0.63 ± 0.02	0.59 ± 0.02	0.089	0.63 ± 0.01	0.58 ± 0.02	0.033

^a^ Continuous variables were expressed as means ± standard error of the mean (SEM), while categorical variables as subject number (percentage distribution). SFA, saturated fatty acid; MUFA, monounsaturated fatty acid; PUFA, polyunsaturated fatty acid.

**Table 2 nutrients-11-00846-t002:** Associations between dietary cholesterol intake and risk for developing abnormal total cholesterol (TC) and low-density lipoprotein-cholesterol (LDL-C) levels ^a^.

Serum Cholesterol Abnormalities	Tertiles of Total Dietary Cholesterol Intake (mg/day)			*p* Value for Trend ^b^
	T1 <155.03	T2 155.03 to 277.51	T3 >277.51	
TC ≥200 mg/dL, *n* (%)	1646 (43.7)	1478 (39.2)	1596 (42.3)	
Crude OR (95% CI)	1	0.863 (0.779–0.957)	0.988 (0.888–1.100)	0.796
Adjusted OR (95% CI) ^c^	1	0.990 (0.897–1.116)	1.153 (0.995–1.337)	0.028
Adjusted OR (95% CI) ^c,d^	1	0.963 (0.854–1.086)	1.104 (0.952–1.282)	0.096
LDL-C ≥130 mg/dL, *n* (%)	1349 (35.8)	1173 (31.1)	1230 (32.6)	
Crude OR (95% CI)	1	0.846 (0.757–0.944)	0.901 (0.808–1.004)	0.139
Adjusted OR (95% CI) ^c^	1	1.037 (0.912–1.179)	1.186 (1.019–1.382)	0.018
Adjusted OR (95% CI) ^c,d^	1	0.991 (0.871–1.128)	1.110 (0.951–1.296)	0.120

^a^ Data were presented as odds ratio and 95% confidence intervals. ^b^ Estimated *p* value for a linear trend was based on linear scores derived from the medians of the tertiles of dietary cholesterol intake among all participants. ^c^ Adjusted for age, sex, BMI, physical activity, education level, drinking, carbohydrate, and protein. ^d^ Additionally adjusted for SFA.

**Table 3 nutrients-11-00846-t003:** Associations between consumption of food sources of dietary cholesterol and the risk for abnormalities in total cholesterol level ^a^.

Serum Cholesterol Abnormalities	Tertiles of Food Consumption (servings/week)			*p* Value for Trend ^b^
	T1	T2	T3	
Red meat, range	<1.70	1.70–3.49	>3.49	
TC ≥200 mg/dL, *n* (%)	1667 (43.9)	1548 (41.6)	1505 (39.7)	
Crude OR (95% CI)	1	0.928 (0.838–1.028)	0.891 (0.802–0.989)	0.039
Adjusted OR (95% CI) ^c,d^	1	1.069 (0.957–1.195)	1.109 (0.983–1.252)	0.111
Processed meat, range	0	0 to 0.58	>0.58	
TC ≥200 mg/dL, *n* (%)	2068 (44.2)	1124 (40.1)	1528 (39.8)	
Crude OR (95% CI)	1	0.871 (0.778–0.975)	0.919 (0.831–1.015)	0.264
Adjusted OR (95% CI) ^c,e^	1	1.047 (0.929–1.181)	1.220 (1.083–1.374)	0.001
White meat, range	<0.23	0.23 to 0.81	>0.81	
TC ≥200 mg/dL, *n* (%)	1719 (42.9)	1590 (42.4)	1411 (39.7)	
Crude OR (95% CI)	1	1.054 (0.947–1.173)	0.925 (0.833–1.027)	0.119
Adjusted OR (95% CI) ^c,d^	1	1.179 (1.052–1.322)	1.088 (0.969–1.222)	0.204
Egg, range	<1.00	1.00 to 3.23	>3.23	
TC ≥200 mg/dL, *n* (%)	1759 (43.0)	1330 (40.9)	1631 (40.8)	
Crude OR (95% CI)	1	0.921 (0.829–1.023)	0.951 (0.856–1.057)	0.505
Adjusted OR (95% CI) ^c^	1	1.006 (0.901–1.123)	1.082 (0.966–1.212)	0.142

^a^ Data were presented as odds ratio (OR) and 95% confidence intervals (95% CI). ^b^ Estimated *p* value for a linear trend was based on linear scores derived from the medians of the tertiles of each food consumption among all participants. ^c^ Adjusted for age, sex, BMI, physical activity, education level, drinking and smoking. ^d^ Additionally adjusted for egg. ^e^ Additionally adjusted for egg and white meat.

**Table 4 nutrients-11-00846-t004:** Associations between consumption of food sources of dietary cholesterol and the risk for abnormalities in low-density lipoprotein-cholesterol level ^a^.

Serum Cholesterol Abnormalities	Tertiles of Food Consumption (servings/week)			*p* Value for Trend ^b^
	T1	T2	T3	
Red meat, range	<1.70	1.70 to 3.49	>3.49	
LDL ≥130 mg/dL, *n* (%)	1375 (36.2)	1227 (33.0)	1150 (30.3)	
Crude OR (95% CI)	1	0.874 (0.789–0.968)	0.809 (0.729–0.898)	<0.001
Adjusted OR (95% CI) ^c,d^	1	1.020 (0.915–1.138)	1.030 (0.914–1.161)	0.656
Processed meat, range	0	0 to 0.58	>0.58	
LDL ≥130 mg/dL, *n* (%)	1684 (36.0)	871 (31.1)	1197 (31.2)	
Crude OR (95% CI)	1	0.848 (0.757–0.949)	0.873 (0.786–0.969)	0.049
Adjusted OR (95% CI) ^c,e^	1	1.030 (0.912–1.163)	1.193 (1.052–1.354)	0.004
White meat, range	<0.23	0.23 to 0.81	>0.81	
LDL ≥130 mg/dL, *n* (%)	1431 (35.7)	1240 (33.1)	1081 (30.4)	
Crude OR (95% CI)	1	0.934 (0.838–1.043)	0.813 (0.725–0.911)	<0.001
Adjusted OR (95% CI) ^c,d^	1	1.052 (0.934–1.184)	0.980 (0.863–1.114)	0.716
Egg, range	<1.00	1.00 to 3.23	>3.23	
LDL ≥130 mg/dL, *n* (%)	1408 (34.5)	1052 (32.4)	1292 (32.5)	
Crude OR (95% CI)	1	0.898 (0.805–1.002)	0.920 (0.825–1.026)	0.232
Adjusted OR (95% CI) ^c^	1	0.995 (0.887–1.117)	1.075 (0.954–1.210)	0.188

^a^ Data were presented as odds ratio and 95% confidence intervals. ^b^ Estimated *p* value for a linear trend was based on linear scores derived from the medians of the tertiles of food consumption among all participants. ^c^ Adjusted for age, sex, BMI, physical activity, education level, drinking, and smoking. ^d^ Additionally adjusted for egg. ^e^ Additionally adjusted for egg and white meat.

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
