# Peer review of "Association between Dietary Cholesterol and Their Food Sources and Risk for Hypercholesterolemia: The 2012–2016 Korea National Health and Nutrition Examination Survey"

_nutrients, 2019, doi:10.3390/nu11040846_

Reviewer 1 Report

Dongjoo Cha and Yongsoon Park have addressed the critical issue regarding the effect of SFA on cardiovascular disease in their research article titled “Association between dietary cholesterol and their food sources and risk for hypercholesterolemia: the 2012-2016 Korea National Health and Nutrition Examination Survey.” This article is well written. The authors have shown the harmful effects of SFA consumption, how it alters the TC and LDL levels in the patients.

No comments regarding the scientific findings in this article,

Author Response

We appreciate your comments.

Reviewer 2 Report

This large scale study describes the association between diet compositiona nd elevated total cholesterol and LDL cholesterol in a large scale Korean cohort.

As such the results showing that the effect of cholesterol intake on TC and LDL-C is abrogated after adjusting for SFA is of interest.

As it stands there are several major issues that need to be addressed by the authors:

1 the association of diet composition with ELEVATED TC and ELEVATED LDL-C is tested.  However, the association with TC and LDL-C as CONTINUOUS variables is at least as relevant. Therefore it is recommended to perform multivariable linear regression analsysis with changes in TC and LDL-C  as continuous variables expressed per MG/DL change or per 1 SD change.

2 it is important to show the indirect effect in the mediation analysis as proportion of total. This holds true for both TC and LDL-C.

Minor:

1 show TC and LDL-C also per mmol/l.

2 adapt the strength and limitation paragraph in so far that the findings are not applicable to other non-Asian populations

3 discuss the weakness of the dietary intake questionnaires.

23 discuss the lack of data on transfatty acids

Author Response

Comments and Suggestions for Authors

This large scale study describes the association between diet composition and elevated total cholesterol and LDL cholesterol in a large scale Korean cohort.

As such the results showing that the effect of cholesterol intake on TC and LDL-C is abrogated after adjusting for SFA is of interest.

Response: We appreciate your comments.

As it stands there are several major issues that need to be addressed by the authors:

1 the association of diet composition with ELEVATED TC and ELEVATED LDL-C is tested. However, the association with TC and LDL-C as CONTINUOUS variables is at least as relevant. Therefore, it is recommended to perform multivariable linear regression analysis with changes in TC and LDL-C as continuous variables expressed per MG/DL change or per 1 SD change.

Response: Thank you for your comments. In our study, there were no significant linear relationships between diet composition and serum levels of TC and LDL-C as shown on the following Table. It is impossible to perform multivariable linear regression analysis in the study. Therefore, we performed multivariable logistic regression to investigate the association of diet composition with the risk of abnormalities in TC and LDL-C.

Table. Associations between dietary cholesterol and their food sources and serum levels of TC and LDL-Ca

TC (mg/dL)

LDL-C (mg/dL)

β

95% CI

p   value

β

95% CI

p   value

Dietary cholesterol (mg/d)b

0.002

(-0.005,   0.008)

0.630

0.000

(-0.006,   0.005)

0.872

Food (servings/week)

Red meatc,d

0.092

(-0.221,   0.405)

0.564

0.074

(-0.209,   0.357)

0.608

Processed meatc,e

0.560

(-0.382,   1.502)

0.244

0.694

(-0.097,   1.484)

0.085

White meatc,d

0.841

(-0.207,   1.889)

0.116

0.482

(-0.429,   1.394)

0.300

Eggc

0.217

(-0.127,   0.560)

0.217

0.095

(-0.207,   0.398)

0.537

aData were presented as β-coefficients and 95% confidence intervals.

bAdjusted for age, sex, BMI, physical activity, education level, drinking and percent of energy from carbohydrate, protein and SFA

cAdjusted for age, sex, BMI, physical activity, education level, drinking and smoking

dAdditionally adjusted for egg

eAdditionally adjusted for egg and white meat

2 it is important to show the indirect effect in the mediation analysis as proportion of total. This holds true for both TC and LDL-C.

Response: We agree with your opinion. We included the sentence in line 123-125 as follows, “In the case of significant indirect effects the proportion of the total effect that was mediated was calculated using the following formula, (indirect effect/total effect) × 100.” and 149-150 as follows, “but SFA contributed to significant indirect effects of 37% and 58% on the relationship between dietary cholesterol and serum levels of TC and LDL-C, respectively.”

Minor:

1 show TC and LDL-C also per mmol/l.

Response: Thank you for your comment. We also showed TC and LDL-C as per mmol/l in line 88.

2 adapt the strength and limitation paragraph in so far that the findings are not applicable to other non-Asian populations

Response: Thank you for your comment. We revised line 243-244 as follows, “First, the findings are not applicable to other non-Asian populations.”

3 discuss the weakness of the dietary intake questionnaires.

Response: Thank you for your opinion. We included the sentence in line 251 as follows, “FFQ is restricted to listed food items and the recall bias of participants might affect the dietary assessment

23 discuss the lack of data on trans fatty acids

Response: Thank you for comments. We included the sentence in line 247-250 as follows, “Fourth, trans fatty acid is one of the major determinants on the level of blood cholesterol. However, due to the missing information regarding the trans fatty acid in database of KNHANES, we could not include effect of trans fatty acid on the associations between dietary factors and serum cholesterol in the present study.

Round  2

Reviewer 2 Report

The revised version of this manuscript has been improved. However there are several isuues that need to be considered in a further revised version of the manuscript:

1 it is suggested that the indirect effect of sfa on tc and ldl-c is statistically significant while the direct effect is not. However numericaaly the direct effect is larger than the indirect effect. Is needs a clear explanation! 

2 while this paper was in the revision period, an interesting paper on dietary cholesterol intake and lipod traits has appeard in JAMA. This peper needs to be included and discussed in the re-revised manuscript.

Author Response

Comments and Suggestions for Authors

The revised version of this manuscript has been improved. However, there are several issues that need to be considered in a further revised version of the manuscript:

1 it is suggested that the indirect effect of sfa on tc and ldl-c is statistically significant while the direct effect is not. However, numerically the direct effect is larger than the indirect effect. Is needs a clear explanation! 

Response: Thank you for your comments. In the case of TC, we found that the numerical value of direct effect (0.0036) was larger than that of indirect effect (0.0021), but the direct effect of 95% CI (-0.0018, 0.0091) was not statistically different from zero. Preacher & Hayes (Behavior research methods, instruments, & computers 2004, 36, 717-731) reported that not numerical value but 95% CI was meaningful.

2 while this paper was in the revision period, an interesting paper on dietary cholesterol intake and lipid traits has appeared in JAMA. This paper needs to be included and discussed in the re-revised manuscript.

Response: Thank you for your suggestion. We included the sentence in line 218–224 as follows, “A recent pooled analysis of 6 prospective cohort studies reported that the associations of dietary cholesterol or egg consumption with incident CVD and mortality were not significant among American adults [42]. However, each additional 300 mg/day of dietary cholesterol consumed or each additional half an egg/day consumed was significantly associated with higher risk of incident CVD and mortality [42]. In the present study, the average intake of cholesterol was 254 mg/day, lower than mean dietary cholesterol consumption of American, and only 28% of population consumed > 300 mg/day.”